# 2D-GaN/AlN Multiple Quantum Disks/Quantum Well Heterostructures for High-Power Electron-Beam Pumped UVC Emitters

**DOI:** 10.3390/nano13061077

**Published:** 2023-03-16

**Authors:** Valentin Jmerik, Dmitrii Nechaev, Alexey Semenov, Eugenii Evropeitsev, Tatiana Shubina, Alexey Toropov, Maria Yagovkina, Prokhor Alekseev, Bogdan Borodin, Kseniya Orekhova, Vladimir Kozlovsky, Mikhail Zverev, Nikita Gamov, Tao Wang, Xinqiang Wang, Markus Pristovsek, Hiroshi Amano, Sergey Ivanov

**Affiliations:** 1Ioffe Institute, 26 Politekhnicheskaya, Saint Petersburg 194021, Russia; 2Lebedev Physical Institute, Leninsky Avenue 53, Moscow 119991, Russia; 3State Key Laboratory for Mesoscopic Physics and Frontiers Science Center for Nanooptoelectronics, School of Physics, Peking University, Beijing 100871, China; 4Institute of Materials and Systems for Sustainability (IMaSS), Nagoya University, Chikusa-Ku, Furo-Cho, Nagoya 464-8601, Japan

**Keywords:** monolayer thick GaN/AlN multiple quantum wells, III-nitrides, carrier localization, plasma-assisted molecular beam epitaxy, electron-beam pumped UVC emitters

## Abstract

This article describes GaN/AlN heterostructures for ultraviolet-C (UVC) emitters with multiple (up to 400 periods) two-dimensional (2D)-quantum disk/quantum well structures with the same GaN nominal thicknesses of 1.5 and 16 ML-thick AlN barrier layers, which were grown by plasma-assisted molecular-beam epitaxy in a wide range of gallium and activated nitrogen flux ratios (Ga/N_2_*) on *c*-sapphire substrates. An increase in the Ga/N_2_* ratio from 1.1 to 2.2 made it possible to change the 2D-topography of the structures due to a transition from the mixed spiral and 2D-nucleation growth to a purely spiral growth. As a result, the emission energy (wavelength) could be varied from 5.21 eV (238 nm) to 4.68 eV (265 nm) owing to the correspondingly increased carrier localization energy. Using electron-beam pumping with a maximum pulse current of 2 A at an electron energy of 12.5 keV, a maximum output optical power of 50 W was achieved for the 265 nm structure, while the structure emitting at 238 nm demonstrated a power of 10 W.

## 1. Introduction

Semiconductor ultraviolet-C (UVC) emitters with operating wavelengths in the λ = 210–280 nm range based on low-dimensional (Al,Ga)N heterostructures are under extensive development for new non-toxic (Hg-free) optical disinfection systems, optical spectroscopy, and communication systems, etc. [1,2,3]. The main challenges for these emitters are to increase their output optical power and expand the operating spectral range in the short-wave subrange below 250 nm, which has a number of interesting applications, namely, in vivo non-cancerogenic disinfection means, resonant Raman spectroscopy, etc. However, UVC LEDs at preset have a relatively low external quantum efficiency (EQE) with a maximum value of 15% and less than 5% in commercial devices [4,5,6]; this limits UVC LED output powers. To date, the reported maximum value is 150 mW for a single-crystal diode emitting at λ = 265 nm, grown on a bulk AlN substrate [7]. Moreover, EQE and output power dramatically decreased with decreasing operating wavelengths, e.g., UVC LEDs with λ~232 nm demonstrated a maximum output power of 1.73 mW at 100 mA under a pulsed-mode operation with a maximum EQE of 0.35% [8].

It is generally accepted that the deterioration of the *p*-type doping of AlGaN layers with an increasing Al content is one of the most urgent problems of UVC LEDs. Therefore, it is of great interest to develop alternative electron-beam (e-beam) pumped UVC emitters (also called UVC-light-source tubes) based on (Al,Ga)N-based heterostructures with multiple quantum wells (MQWs) without *p*-type doped layers [9,10,11,12,13,14,15,16,17]. Moreover, these UVC emitters have demonstrated a uniquely high output optical power-up to the Watt range.

The results above were obtained for light-emitting devices with conventional (Al,Ga)N MQW structures with a typical well width of 1–3 nm, possessing several additional factors that contribute to a low EQE. These are the decay of internal quantum efficiency due to the quantum confinement Stark effect (QCSE) and reduction in *c*-plane light-extraction efficiency due to switching from transverse electric (TE) to transverse magnetic (TM) polarization modes for output radiation [18,19,20,21].

One of the most efficient ways to mitigate these problems is to develop GaN/AlN MQW structures with an ultimate thickness of one or a few monolayers (MLs) (1 ML~0.249 nm (0.259 nm) for AlN(GaN), respectively). Such *N* × {GaN_m_/AlN_n_} MQW structures, where *N* is the number of QWs (periods) and *m* and *n* are the nominal thicknesses of well and barrier layers in MLs, can be grown using both plasma-assisted molecular-beam epitaxy (PA MBE) [22,23,24] or metalorganic chemical vapor deposition (MOCVD) [25,26]. Moreover, eliminating the limiting electrical injection via a *pn*-junction allows an increase in the QW number up to several hundreds to provide a relatively high output optical power of about 11 W at 240 nm from e-beam pumped MQW structures [24]. 

In our previous works on such GaN/AlN MQW structures, the QW thickness was varied in the range of 0.75–7 ML to study the effect on the emission wavelength and output power of UV radiation. It was shown that a decrease in the QW thickness to 1–2 ML resulted in a blue shift of the radiation wavelength to the UVC spectral range of 230–260 nm. Moreover, theoretical and experimental studies of the electronic structure of the ML-thick QWs confirmed the excitonic character of luminescence, which results in a high internal quantum yield (IQY) of up to 75% at room temperature [27,28]. Our previously demonstrated structures employed ultrathin multiple QWs (thickness < 2 ML) formed by 2D-growth owing to the metal-enriched growth conditions. These QWs did not show a transition to a three-dimensional (3D) Stranski–Krastanov (SK) growth, which is commonly used to form an array of quantum dots (QDs) of a narrower-gap material in a wider-gap matrix, e.g., for GaN/AlN heterostructures [29,30,31].

A tunable sub-230 nm deep-UVC emission was demonstrated by Jena’s group, which used PA MBE in a modified SK growth mode to form 2 ML GaN quantum dots/disks via the thermal annealing of the 2 ML-thick GaN QWs sandwiched between AlN barriers [32]. Then, this group developed an alternative approach to achieve tunable sub-230 nm radiation based on the SK growth of GaN/AlN dots/disks under nitrogen-rich conditions [33]. However, the growth kinetics of ultrathin GaN/AlN MQWs at different Ga to activated nitrogen flux ratios (Ga/N_2_*) and the relationship between the internal morphology of QWS and radiative properties remain poorly studied areas.

This paper presents the results of a comprehensive study of GaN_1.5_/AlN_16_ MQWs with the same nominal well width of 1.5 ML, grown by PA MBE under different stoichiometric conditions from nitrogen to gallium-rich, while the growth parameters of 16 ML-thick AlN barrier layers were fixed. Two types of AlN/*c*-Al_2_O_3_ templates were used to study the effect of the template on the characteristics of the QW morphology, which governs the degree of charge-carrier localization and the respective change in the wavelength of the UVC radiation. Varying the stoichiometric growth conditions makes it possible to tune the emission wavelength in the 238–265 nm range and achieve an output power of several tens of Watts.

## 2. Samples and Characterization Methods

All samples were grown using a Compact21T PA MBE setup (Riber, Bezons, France) with an HD-25 plasma source (Oxford Appl. Res. Ltd., Oxfordshire, UK) on two types of AlN/*c*-Al_2_O_3_ templates. Most of the samples were grown on single 1.5 µm-thick AlN templates fabricated using PA MBE, as previously described [34]. This type of template was designated as an S-type. Several samples were grown on double (composite) AlN templates (designated as D_i_-type with *i* = 1, 2) consisting of an initial MOCVD-grown layer with a thickness of 0.7 μm and a subsequent 0.9 μm-thick AlN layer grown using two different PA MBE growth schemes. In particular, D_1_-type templates were grown under Al-rich conditions by using metal-modulated epitaxy (MME), leading to a continuous 2D-surface of AlN layers, as described in Ref. [35]. D_2_-type templates were also grown mainly using the same Al-rich MME conditions but with a short-term transition to nitrogen-rich conditions, resulting in a reversible 2D–3D–2D growth-mode transition for these templates. 

The MQW structures were grown at the constant substrate temperature of 690 °C; and plasma-activated nitrogen flux N_2_* = 0.47 ± 0.02 ML s^−1^, which was provided by the plasma source excited by an RF-power of 150 W and a neutral nitrogen flow of 4 sccm. Most of the MQW samples were grown with the same nominal QW thickness of 1.5 ML (only 1 had a thickness of 1.2 ML), but at various Ga/N_2_* flux ratios by changing the Ga flux, as shown in Figure 1. In addition, MQW structures with wells 1 and 2 ML thick were grown. The samples were named accordingly: Q (nominal well thickness)—(Ga/N_2_* ratio)—(type of AlN template: S or D_i_ *i* = 1, 2).

The 16 ML-thick AlN barrier layers were grown under slightly Al-enriched conditions with a flux ratio of Al/N_2_* = 1.1. Please note that the growth conditions of the AlN barrier layers were changed compared to our previous work [24], namely, two MLs of aluminum were deposited just before the start of the growth of AlN barrier layers, as shown in Figure 1.

Most of the MQW structures consisted of 400 periods, but one had 100 periods (designated as 100 × Q1.5-2.2-S). In addition, a test AlN layer was grown on an S-type AlN template with 400 growth interruptions, during which the surface was exposed to nitrogen flow similar to the MQW structures, but without the GaN QW growth stage.

All stages of the PA MBE growth of AlN templates and ML-GaN/AlN MQW structures were monitored using a reflection high-energy electron diffraction (RHEED) (Staib Instrumente GmbH, Langenbach, Germany), infrared pyrometer Mikron M680 (Mikron Infrared, Inc., Oakland, NJ, USA), as well as home-made laser reflectometry and multi-beam optical stress sensor (MOSS) [36].

The postgrowth studies of the structures were conducted using an atomic force microscope (AFM) Ntegra AURA (NT-MDT, Moscow, Russia) in a semicontact mode using silicon HA-NC probes (NT-MDT) with a tip apex radius of 10 nm and a force constant of 3.5 N/m. In addition, the structures were studied by high-resolution X-ray diffractometry and reflectometry (HRXRD, XRR) using a D8DISCOVER, (Bruker AXS, Karlsruhe, Germany) diffractometer with a rotation anode (Cu Kα1 radiation). The strains in the structures were determined from reciprocal space maps (RSMs) for the asymmetric (105) reflection. The average compositions and periods in the GaN/AlN MQW structures were measured by XRD 2θ/ω symmetric (002) scans. In addition, the periods, surface topographies, and morphologies of the internal interfaces in the GaN/AlN MQW structures were studied using the same diffractometer by measuring the X-ray reflectometry (XRR) and ω-curves near the Bragg-like peaks. HRXRD and XRR data were simulated using the Leptos software package version 7.04 from Bruker AXS, Karlsruhe, Germany. Omega curves of small-angle X-ray scattering were simulated within the framework of the Ming model included in the same software [37].

High-angle annular dark-field scanning transmission electron microscopy (HAADF-STEM) images were recorded using a spherical aberration-corrected FEI Titan Cubed Themis G2 300 transmission electron microscope (TEM) operated at 300 kV. The TEM samples were prepared by mechanical polishing and argon ion milling using Gatan PIPS™ Model 691.

The photoluminescence (PL) spectra of the structures were measured with liquid nitrogen (77 K) and room temperature (RT) using the 4th harmonic of a Ti-sapphire laser Mira 900 (Coherent, Santa Clara, CA, USA) with a wavelength of 211 nm as the excitation source.

Two types of home-made electron guns (e-guns) were used to measure the cathodoluminescence (CL) spectra of the structures. Initial studies were conducted using e-guns with thermionic cathodes, which provided continuous wave (cw) e-beams with an energy of 10 keV, a current of 30 nA, and a diameter of 2 µm, or with the energy of 10–30 keV, a 5 µA current, and a diameter of 1 mm. The spectra were measured in the back-scattering geometry of the optical luminescence in respect to the exciting e-beam. Significantly higher-current e-beams could be obtained using a e-gun with a ferroelectric plasma cathode operated in a pulsed mode with a frequency and pulse duration of 1.5 Hz and 0.5 μs, respectively. The maximum e-beam current could reach 2 A at an electron energy of 12.5 keV and a beam diameter of 4 mm. The luminescence was excited by the e-beam from the side of the structure’s surface and measured from the back side of the sapphire substrate.

## 3. Results and Discussion

### 3.1. Growth Kinetics of GaN/AlN MQW Structures

The main aim of this study was the fabrication of ultrathin GaN/AlN QWs with a well thickness of 1.5 ML (0.37 nm), which did not exceed the critical value of the Stranski–Krastanov transition in this heterostructure, reported as 2 ML by different groups [38,39,40,41]. Moreover, these works show that PA MBE can provide a 2D-growth mode of ultra-thin (up to 2 ML) GaN QWs grown in AlN over a wide range of the Ga/N flux ratio (0.8–1.6) and substrate temperature (690–740 °C). In addition, the low growth temperature of about 690 °C prevented the segregation and interdiffusion of Ga atoms at the GaN/AlN heterointerfaces [42]. Meanwhile, this temperature was high enough to provide a sufficient desorption rate of excess Ga atoms (~0.223 ML s^−1^ [35]) from the surface of the growing MQW layers to avoid the formation of metal droplets. A minor excess of Al atoms on the surface of the AlN barrier layers grown under slightly Al-rich growth conditions was consumed when the AlN surfaces were exposed to an N_2_* flux before the growth of each GaN QW. These considerations were used in choosing the PA MBE parameters for the MQW structures, in accordance with the scheme shown in Figure 1.

The absence of 2D–3D transitions during MQW growth was confirmed by the RHEED data presented in Figure 2, which shows continuous streaky RHEED patterns over the growth of one period of the Q1.5-2.2-S structure, marked as A_i_–D_i_ in this figure.

However, the brightness of all RHEED streaks changed during the growth of each layer of the GaN/AlN MQWs, as shown in the RHEED intensity panel in Figure 2 for a specular reflex. These changes exhibited the periodic nature and persisted throughout the growth of 400 MQW periods with a total thickness of 1750 nm. The initial decrease in RHEED intensity during QW growth was mostly due to the excess Ga accumulation on the surface, estimated as 1 ML for the Q1.5-2.2-S structure (see Appendix A); however, an effect of the interaction of electrons with Ga atoms in GaN having a higher atomic number than the Al ones in AlN cannot be neglected. A slight increase in the RHEED streak intensity during the deposition of the Al bilayer supports the latter assumption. A further increase in the RHEED intensity during the AlN barrier-layer growth was presumably governed by the thermal desorption of the accumulated Ga atoms, which segregated on the surface of the growing AlN layer (Al/N_2_* > 1). The estimated Ga desorption rate from pure Ga droplets on the surface of the GaN bulk layer was determined to be about 0.22 ML·s^−1^ at *T*_S_ = 690 °C [35], which was higher than the experimental value derived from the RHEED intensity behavior during AlN growth and illustrated in the bottom diagram in Figure 2. The slowing down of Ga desorption was likely caused by the complex Ga–Al composition of the metal accumulated phase, being proportional to the Ga content in the accumulation layer. The final RHEED intensity recovery occurred during the Al-bilayer consumption stage under the N_2_* flux during growth interruption. The continuous 2D-RHEED patterns with a periodical modulation of their intensities were also observed during growth of MQWs under Ga/N_2_* flux ratios of 1.1 and 0.6.

### 3.2. Stress Evolution during the Growth of GaN/AlN MQWs

Figure 3a shows the (stress × thickness) product variation monitored in situ by MOSS of the three-stage PA MBE growth of an S-type AlN template layer on *c*-Al_2_O_3_, used for the following growth of the Q1.5-2.2-S sample, which yielded a very low average tensile stress value of σ_av_ ~0.15 GPa. All other S-type MQW samples in this study were grown on similar quasi-stress-free AlN/*c*-Al_2_O_3_ templates fabricated under optimized PA MBE growth conditions obtained in our previous experiments [43].

In contrast, much higher values of σ_av_~0.95 and 0.8 GPa were observed during the growth of D_1_- and D_2_-type AlN templates, respectively, as shown in Figure 3c. The higher value was obtained from the MOCVD AlN template grown at a much higher growth temperature of 1280 °C with a considerable contribution from the Hoffman–Nix–Clemens mechanism for tensile stress generation due to growth grain coalescence [44]. The lower σ_av_ value of the latter result is related to the short-term transition to the 3D-growth mode, which apparently results in a decrease in the incremental tensile stress in the D_2_-type AlN template. Other implication of this transition are discussed later.

The MQW structures grown on S- and D-type templates exhibited opposite stresses. The MQWs presented a negative (compressive) stress sign on S-templates, which corresponds to the expected crystallographic mismatch between the AlN template and the GaN/AlN MQW structure, while the MQWs on D-substrates only showed tensile stress. Meanwhile, the observed stress values in MQW structures on the S-type templates were significantly lower than the nominal value of −0.9 GPa calculated for the stress in the MQW GaN/AlN structure with an average Al mole content of 91 mol% on the AlN layer, as described in Appendix A. Moreover, only structures grown under Ga-enriched conditions demonstrated an approximately constant compressive stress of (0.4–0.5) GPa, while structure Q1.5-0.6-S grown under N-rich conditions revealed a gradual decrease in compressive stress. It can be assumed that this decrease was due to its 3D grain topography, leading to the appearance of tensile stress, which compensated for compressive misfit stress, as discussed in our previous work [43]. 

As for the D-type templates, a much higher positive (tensile)-stress level of up to 1.45 GPa was observed in the MQW structure grown on the D_1_-type template (blue line in Figure 3d). A sharp stress-relaxation effect occurred at the thickness of 0.75 μm. An average stress level of about 1.2 GPa can be calculated in the D_1_ template and part of the MQW structure, having a total thickness of 1.75 µm, before relaxation. These values correspond to the typical critical (stress × thickness) product for the onset of cracking as observed by Hearne et al. [45] for the AlGaN/GaN heterostructure with similar tensile stress values. Consequently, the Q1.5-1.1-D_2_ structure with a lower average tensile stress value of ~0.6 GPa in the template and the MQW structure showed no signs of cracking throughout the whole growth process up to the total thickness of 2.75 μm.

### 3.3. Structural Properties of GaN/AlN MQWs Structures

#### 3.3.1. Surface Topographies of MQW Structures

Figure 4 shows the surface topographies of layer and MQW structures measured by AFM for a variety of conditions. Figure 4a,b show the surface of the AlN test layer grown without GaN MQWs, which has a low Root Mean Square (RMS) roughness of about 2 ML over a scanning area of up to 5 × 5 µm^2^ (Table 1).

Despite the low rms roughness, this layer did not show signs of a spiral growth mode with a terrace-stepped surface topology, which is usually observed for AlN buffer layers grown at higher substrate temperatures (780/850 °C) using metal-modulated epitaxy [34]. The AFM images in Figure 4a,b show only faint evidence of local atomic steps. This can be explained by the relatively low growth temperature of ~690 °C, which suppresses spiral growth by reducing the surface mobility of adatoms.

The Q1.2-0.6-S structure with QWs grown under nitrogen-rich conditions had a much worse morphology with the highest RMS roughness of up to several nm, as shown in Figure 4c,d and Table 1. This corresponds to the well-known phenomenon of the limited surface mobility of adatoms under such conditions, which promotes the 3D growth of III-nitrides [46,47]. Even though these conditions were only used for QW growth, the subsequent growth of the AlN barrier layer in Al-enriched conditions could not restore the surface’s smoothness due to limited Al mobility at 690 °C. Moreover, the minimum compressive stress in this structure (see above, Section 3.2) corresponded to 3D topology, which promotes the development of tensile stress in such structures due to the cohesive interaction between the individual grains [43].

In contrast, the surfaces of the rest of the structures studied with QWs grown under metal (Ga)-rich conditions consisted of densely packed hexagonal hillocks with a terrace-stepped surface topography, as shown in Figure 4e–j. The step-spacing at the spiral sidewalls would be limited by the step-layer growth mechanism according to the classical model of Burton, Cabrera, and Frank (BCF) [48]. This indicates a relatively long diffusion length of adatoms exceeding the terrace widths of 30–40 nm under Ga-rich condition. The structure grown under Ga-rich conditions (Figure 4g,h) exhibited steps with a height of one *c*-lattice constant (=5.18 Å, i.e., 2 ML). Similarly, a thinner structure of 100 × Q1.5-2.2-S grown under the same conditions presented a very similar surface topography (Figure 4i,j). It should be noted that the typical features of 2 ML-high steps and constant surface topography vs. growth time were observed by Natali et al. [49], who studied GaN/AlGaN QW structures grown by PA MBE at various conditions and thicknesses.

The Q1.5-1.1-S structure grown under reduced Ga-rich conditions demonstrated a less pronounced spiral morphology with a distinct formation of individual islands (Figure 4e,f). This can be explained both by the deficiency of Ga atoms during the growth of the QWs and the lower accumulation of these atoms, which would create a surfactant effect during the subsequent growth initiation of the AlN barrier layers. As a result, the diffusion length of adatoms during the growth of both QWs and barrier layers reduced, increasing the probability of 2D nucleation on the terraces.

Moreover, the step’s height was only one ML (*c*/2) under these conditions, which contributed to less-pronounced spiral shapes as compared to higher Ga-rich conditions, as was also observed in Ref. [49]. We called this the “mixed-growth mode” because it features both step-flow and 2D-nucleation mechanisms during the growth stage. The mixed mode for the Q1.5-1.1-S structure provided the lowest RMS roughness value of about 0.35 nm, while the Q1.5-2.2-S structure with dominating spiral growths had the higher RMS roughness of 0.79 nm in the same AFM scanning area of 1 × 1 µm^2^; see Table 1.

It is also worth noting that the RMS roughness of all MQW structures grown on the S-type template depended relatively weakly on the AFM scan area of up to 15 × 15 µm^2^, as shown in Table 1, which was due to their spiral sizes being smaller than the scan area. Furthermore, the RMS roughnesses described above were significantly reduced compared to those in our previous work [24], which had typical RMS values around 2 nm, even for scan areas of 1 × 1 µm^2^. The smoother layers most likely appeared due to the predeposition of 2 ML of Al before starting the growth of AlN barrier layers (Figure 1).

Thus, the improvement in the surface morphology with the QWs grown under Ga-rich conditions is apparently related to the excess of Ga and Al adatoms on the surface during the growth period. As discussed in Section 3.1, excess Ga does not instantly evaporate from the surface of the AlN barrier layer at the growth temperature of 690 °C, and, rather, acts as a surfactant to increase the surface mobility of Al adatoms and smoothen the surface. The degradation of the surface of the Q1.1-0.6-S structure with MQWs grown under nitrogen-rich conditions therefore occurred due to the absence of the Ga surfactant during the subsequent growth of the AlN barrier layers.

Figure 5 shows AFM images of the Q1.5-1.1-D_1_ and Q1.5-2.2-D_2_ MQW structures grown under the same metal-enriched conditions as the structures on the S-type templates described above. Both showed a significantly lower density of nanohillocks located at the tips of the growth spirals of around (3–7) × 10^8^ cm^−2^ (in Figure 5) compared to the analogous hillocks formed on S-type templates of (2–5) × 10^9^ cm^−2^ (in Figure 4). This difference was most likely due to the different densities of screw-type threading dislocations in the templates, which serve as centers of spiral growth. Indeed, the X-ray diffraction analysis of the symmetric (0002) AlN reflection in the samples determined a full width at half maximum (FWHM) of 47 arcsec for the D-type AlN template, while the FWHM was 450 arcsec for the S-type template.

Moreover, spiral growth dominated in both structures. The higher density of the nanohillocks was observed in the Q1.5-1.1-D_2_ structure compared to Q1.5-2.2-D_1_, which was attributed to the lower surface mobility of adatoms at the lower Ga/N_2_* ratio in the Q1.5-1.1-D_2_ structure (top row). Moreover, Figure 5d for the Q1.5-2.2-D_1_ structure shows extended local perturbations in a large scanning area of 20 × 20 μm^2^ of the surface, which leads to a high RMS roughness value of about 10 nm for this structure (Table 1). These perturbations are traces of cracks in this structure, as observed in the in situ stress monitoring discussed in Section 3.2. The SEM image of the structure, clearly showing cracks, can be found in Appendix A.

#### 3.3.2. Studies of GaN/AlN MQW Structures Using X-ray Diffraction (XRD) and Reflectometry (XRR)

The RSMs of several MQW structures are shown in Appendix A, which demonstrate the vertical alignment of AlN and MQW peaks, indicating the strained growth of the structures. The simulation in the coherent approximation of the θ/2ω scans of symmetric (002) XRD reflex shows that the average Al composition differs slightly from the nominal values with a typical deviation of up to 6%. From the data presented in Table 2, it follows that the closest match (the difference between the simulated and nominal values of ~3%) can be observed in the relatively thin structure, 100 × Q1.5-2.2-S, while in the thicker one, for Q1.5-2.2-S grown under the same Ga-rich conditions, this difference exceeds 5%. Approximately the same difference was observed for all other structures.

Figure 6a shows θ/2θ scans of the XRR curves of the MQW structures presented in Table 2, where the positions of the Bragg-like peaks *I*, *II*, and *III* determine the MQW period. The results are also included in Table 2. They reveal a difference between the simulated and nominal values of the period in most structures at a level not exceeding ~1 ML with a minimum difference of <2% for the thinnest structure: 100 × Q1.5-2.0-S. In both cases, the lower Ga-N binding energy compared to Al-N played a central role, which can lead to a significant increase in the rate of the Ga-Al-exchange reaction. Both metal-enriched growth conditions and compressive stresses are known to increase the rate of this reaction [50,51]. However, the extremely low growth temperatures (below 700 °C) of the GaN/AlN MQW structures used in this study should kinetically limit the occurrence of both reactions described above [42].

In addition, the greatest deviations between the nominal and simulated values of the MQW period and average composition were observed in the structure with the worst surface topography (Q1.2-0.6-S) due to its growth under nitrogen-enriched conditions. Therefore, these inconsistencies can be associated with their relatively high roughness value exceeding 1 nm, which significantly exceeded their nominal fractional thickness value of 1.5 ML (0.389 nm). The difficulty in achieving high accuracy in modeling such ML-thick QWs was noted by Fewster et al. [52], Jenichen et al. [53], and in our previous work [24]. The reasons for the observed discrepancies presented in the QW thickness values are addressed elsewhere.

In this work, we focused on the evaluation of the surface topography and internal interfaces in ML-thick GaN/AlN MQW structures using measurements and the modeling of the specular reflection and diffuse scattering of X-rays at grazing incidence. The measured x-ray specular reflection curves in Figure 6a exhibit different decay rates, which makes it possible to estimate the surface RMS roughnesses of the MQW structures, which are presented in the last column of Table 2. These roughnesses agree with the results obtained using AFM (Table 1). Some of the differences in the roughnesses were also due to the different areas analyzed: very local for the AFM and up to a few square millimeters for XRR.

Figure 6b,c show the results of the measurements and simulations of the ω scans of the first Bragg reflections in the different MQW structures. Most of these curves demonstrate both specular and diffuse components; however, the former is completely absent in the Q1.2-0.6-S structure with MQWs grown under nitrogen-enriched conditions. This not only corresponds to the greatest roughness of its surface as measured by both AFM (Figure 4c,d) and the specular reflection 2θ/ω scan (Figure 6a), but also indicates a high intermixing at the internal interfaces in this heterostructure. In contrast, the highest intensity of the specular reflection and the lowest intensity of the diffuse scattering component was observed in structure Q1.5-1.1-D_2_ grown under slightly metal-enriched conditions (Figure 6b,c), with a total (including interface) roughness of 0.3 nm (Table 3). This agrees with the AFM data showing the lowest RMS roughness of ~0.30 nm for this structure due to the spiral growth with a relatively low density of spirals. A somewhat higher diffuse-scattering component and a weaker specular refection were observed in the Q1.5-1.1-S structure with a roughness of 0.6 nm, which, in accordance with AFM data, was grown in the mixed-spiral and 2D-nucleation growth mechanisms. XRR indicated a slightly greater roughness of 0.8 nm for structure Q1.5-2.2-S, which was grown under strongly metal-rich conditions. Thus, the smoothest surface topology and sharpest QW interfaces for ML-thick GaN/AlN MQW structures can be achieved through PA MBE growth in slightly Ga-rich conditions (Ga/N_2_* = 1.1) on the D- or S-type templates, resulting in spiral step-flow or combined (spiral step-flow and 2D-nucleation) growth mechanisms, respectively.

The roughness in Table 3 again correlates with the AFM RMS roughness for these structures. Additionally, a strong correlation of the inner interfaces at the top of the AlN barrier surface was expected due to the low growth temperature and the limited mobility of the Al and Ga atoms.

The simulation of the diffuse reflection provided unique information for the roughness of internal interfaces in the MQWs. Moreover, the diffuse scattering analysis allows the determination of the lateral correlation lengths (Table 3), which approximately corresponded to the characteristic diameters of GaN islands in the MQW structures. The higher values of this parameter (>200 nm) for the structures grown under less metal-rich conditions indicated their relatively high degree of planarity. Moreover, these structures exhibited a lower Hurst parameter, meaning «jagged» interfaces and rather weak waviness was observed via their AFM images. In contrast, the highly metal-rich Q1.5-2.2-S showed a larger Hurst parameter, indicating «smooth» hills and valleys (i.e., increased short-range variations, together with short correlation lengths). On the other hand, only this structure had a vertical correlation of roughness with a correlation length of 4 nm, which is approximately equal to the QW period of the structure. It is likely that the strong local variation in the GaN layer provided enough strain to enhance the formation of a similar structure in the subsequent layer.

#### 3.3.3. HAADF STEM Study of the GaN/AlN MQW Structure

Figure 7 shows cross-section HAADF-STEM images at various magnifications of the Ga-rich structure Q1.5-2.2-S. First, the periodicity of QWs in this structure with an average period of 4.180 nm is close to 4.00 ± 0.08 nm from the XRR, which is a good result for STEM. Moreover, QWs are less than 2 ML high, which confirm the 2D-growth of GaN QWs on the stepped surfaces of AlN barrier layers with an equilibrium step height of 2 MLs, as found in the AFM images in Figure 4g,h. The upper and lower QW interfaces are identical and do not exhibit different degrees of sharpness in the growth direction, which is usually observed in structures grown at higher temperatures (>720 °C), and then leads to a diffuse upper interface due to the segregation of Ga atoms [42]. Thus, HAADF-STEM confirms the suppression of Ga segregation at the low growth temperature of about 690 °C.

These HAADF STEM images show a non-uniform spatial distribution in the lateral direction, visible as regions of different levels of brightness. Despite the continuous 1 ML-thick contrast throughout the QW plane, there exist obvious local thickness broadenings up to 2–3 ML. To qualitatively characterize these contrast variations in the QWs’s thickness, the line profiles of reciprocal local brightness along the QWs selected by white frames were plotted below each HAADF STEM image. This integrated brightness in each position corresponds to the mean GaN content in these atomic rows. Hence, this profile presents the upper limit of the fluctuations of the effective energy bandgap of the GaN/AlN QWs. The characteristic lateral size of the GaN content fluctuations is defined as the average distance between the adjacent minima as about 10 nm, although it decreases to 5–6 nm in some areas. Since STEM lacks information about the GaN content distribution in the normal direction, the fluctuation size is rather an upper limit and may vary even more, given a similar variation in the normal direction. (The STEM images are too small to compare it with the correlation length derived from the diffuse XRR scattering analysis).

Thus, STEM finds that the GaN QWs with a nominal thickness of 1.5 ML are the non-uniform arrays of regions that may be called quantum disks (QDKs), where a local thickness varies from 1 to 2 ML. This morphology is likely caused by the growth of GaN on AlN terraces with an equilibrium atomic step height of 2 ML. Ultimately, thin QDKs and the possibility of elastic stress relaxation due to the non-continuous QW plane topography on a stepped surface exclude the transition from 2D- to 3D-growth modes. These QDKs can efficiently localize excited carriers even at high temperatures due to the strong energy-gap differences between GaN and AlN. Thus, the method may be considered as an alternative to the traditional Stranski–Krastanov growth mechanism of conventional GaN/AlN quantum dots with greater vertical thickness values, arising after the transition from 2D- to 3D-growth modes of GaN wetting layers at its critical thickness value. 

A similar formation mechanism of isolated clusters at the terrace edges was described by Ploog and Brandt [54] when studying the sub-monolayer MBE growth of coherent InAs insertions with a nominal fractional thickness (0.8 ML) in a GaAs matrix. Using XRD and TEM, they showed the formation of InAs clusters with subnanometer sizes in the vertical and lateral directions on a terraced vicinal surface of GaAs as a result of step-edge nucleation of In adatoms. 

In 2001, Krestnikov et al. [55] reviewed the growth and properties of ultrathin sub-ML narrow-band-gap insertions in wide-band-gap matrices. By examining these structures in various III-V, II-VI, and III-N materials, they demonstrated the formation of arrays of islands that act as locally formed QW insertions in the case of a relatively large lateral size, or exhibited QD properties with a decrease in this size. The formation of precursor, flat, 2D islands with a radius of ~10 nm was experimentally observed by the Daudin’s group [39] during the PA MBE growth of (0001) GaN layers with a subcritical nominal thickness (<2.25 ML) on AlN templates with 30 nm-wide terraces and spiral hillocks. Since this group had nitrogen-rich conditions and a relatively high growth temperature (730 °C), their results cannot be directly transferred to our work. However, the observation of nanoclusters at a subcritical thicknesses level of GaN before the Stranski–Krastanov transition indicates the possibility of a similar formation mechanism.

### 3.4. Studies of the Optical Properties of GaN/AlN MQW Structures 

#### 3.4.1. Photoluminescence Spectra

Figure 8a shows the PL spectra measured at RT and 77 K for the MQW structures with the same nominal QW thickness (1.5 ML), except one having that of 1.2 ML, and a barrier layer thickness of 16 ML, grown at a Ga/N_2_* flux ratio varying from 0.6 to 2.2. The absolute peak intensities in the spectra were normalized to the maximum PL intensity observed for Q1.5-1.1-D_2_ at 77 K. All spectra exhibited single bands in a wide spectral range from 4.7 to 5.7 eV (220–265 nm) with different intensities and linewidths.

The brightest RT PL spectra with a relatively high *I*_PL_(RT)/*I*_PL_(77) ratio of up to 0.9 were observed in MQW structures grown under Ga-rich conditions. An analysis of the spectral position of these single-peak spectra, located in the energy (wavelength) ranges of 4.7–5.4 eV (230–265 nm), revealed that the structures grown at a maximum Ga/N_2_* ratio of 2.2 exhibited peaks in the long-wavelength part of this range, at 250–265 nm. At the same time, the PL peaks in the structures grown at a lower Ga/N_2_* ratio of 1.1 that were blue-shifted to the short-wavelength range of 230–245 nm. Moreover, the PL peak position of the structures almost linearly depended on the Ga/N_2_* ratio used, as shown in Figure 8b, which is discussed later. It should also be noted that the intensity of the RT spectra significantly decreased as they were shifted to the short-wavelength region upon the transition to less Ga-enriched QW growth conditions (Figure 8a). 

On the other hand, Figure 8c shows two PL spectra for MQW structures with nominal integer QW thicknesses of 1 and 2 ML, which reveal single peaks at significantly different wavelengths of 224 and 255 nm (with an energy difference of 0.70 eV). These results agree with the theoretical calculations of the effective band gap in ML-thick GaN/AlN QW structures [56], as well as the experimental results obtained by Jena’s group for similar QWs [32]. In fact, these spectra roughly outline the spectral boundaries at which the radiation of the studied fractional monolayers can be located.

The behavior of the PL spectra corresponds to changes in the internal morphology of the MQWs when the Ga/N_2_* ratio changes from 2.2 to 1.1. Even if both exhibit a 2D surface, the spiral step-flow growth is more pronounced for the samples grown at Ga/N_2_* = 2.2 due to the formation of a Ga-bilayer with a high adatom mobility (2 ML steps are visible in Figure 4g). Hence, the thickness fluctuation of the QWs would be greater and thus at a higher energy, resulting in a longer wavelength emission. The PL spectra of the Q1.5-2.2-D_1_ structure clearly resembled the spectra of the 2 ML QW in Figure 8c. Therefore, we can assume the formation of local islands with an effective thickness of 2 ML on atomically smooth terraces with a lateral size of 20–30 nm up to the whole width of the terrace. In other words, these fluctuations are much greater than the exciton radius, which cause us to consider the islands as QWs rather than QDKs.

In contrast, the structures grown at a reduced Ga/N_2_* ratio of 1.1 exhibited a transition from pure spiral step-flow growth mode to a mixed-growth mode with reduced adatom mobility, which favors the formation of higher-density QDKs with a thickness of 1 ML and smaller lateral dimensions. The smaller average sizes may lead to the lateral confinement of excitons. As a result, the PL peaks of Q1.5-1.1-S and Q1.5-1.1-D_2_ structures lie at around the 1 ML QW peak position but have a stronger 77 K PL. 

In the nitrogen-rich condition, the formation of disk-shaped GaN regions is suppressed and their sizes are reduced, since their nominal thickness is 1.2 ML, and not 1.5 ML, as in all other structures. Therefore, these MQWs emit power at the highest energy level, and the low signal intensity is both due to reduced confinement as the emissions are close to the AlN energy gap and greater roughness in this structure. 

The observed dependence of the PL maximum spectral position on the growth conditions generally resembled a similar dependence for standard GaN/AlN QDs with a few-nanometers thickness, where this effect was explained by a decrease in the QCSE due to the internal polarization field in the wurtzite heterostructures [18,57]. Numerous theoretical and experimental works also indicated the decrease in/absence of QCSE in ultra-thin (1–2 ML) QWs [25,27,28]. Interestingly, in the standard QDs, the blue shift of the PL peak was accompanied by an increase in the PL intensity due to the improved overlap of the electron and hole wavefunctions in narrower wells. On the contrary, the PL peak intensity in Figure 8a exhibits complex behavior at the blue-shift area, which can reflect its dependence on the morphology and partial filling of the surface with QDKs as well.

The 1–2 ML QDKs observed in this study are more efficient at localizing carriers and limiting their lateral transport to nonradiative recombination sites, such as threading dislocations or point defects. Thus, they strongly increase the internal quantum efficiency as demonstrated by the temperature variable PL.

The optical properties of the ML-thick QW structures were determined by excitons up to RT, as it was predicted theoretically [27] and experimentally demonstrated in our previous work [28]. In this case, some delocalization of carriers from shallow localization islands at relatively low temperatures and, hence, a local change in the potential relief for excitons may explain the narrowing of the PL band at RT as compared to that at 77 K (see Figure 8a). We also highlighted a relatively minor difference between the PL spectra for structures grown on S- and D-type AlN templates. This implies that the characteristic size of charge carrier-localization regions is much smaller than the average distance between the defect centers, and for QDKs the distance to a spiral was of no importance, as was concluded from STEM studies.

QW width fluctuations in the ML range were initially studied in the 1980s for GaAs/AlGaAs QW structures [58,59,60,61]. In these works, the ML fluctuations were much smaller than the nominal well width (about 10–20 nm). As a result, the effects of these fluctuations were weak and manifested at low temperatures in a slight broadening of the PL lines and a small increase in the so-called “Stokes shift” (the energy difference between the adsorption edge and corresponding PL line). A significantly greater spectral broadening due to well-width fluctuations was theoretically and experimentally determined for the III-nitride GaN/AlGaN, InGaN/GaN QW samples [62,63]. This effect was explained by the presence of a strong polarization field in the nitride system.

On the contrary, the results described in this section demonstrate the strong effect of the quantum confinement of the exciton energy in the (1–2) ML-thick GaN/AlN QDKs at a negligibly small QCSE in these structures.

Thus, in this section, we compared the PL spectra of ML-thick GaN/AlN structures grown at different Ga/N_2_* flux ratios, which resulted in different QW growth mechanisms, as shown in 3.1.1. It was found that the position of the PL peaks could be controlled in the spectral range from 230 to 265 nm (4.7–5.4 eV) by increasing the Ga/N_2_* flux ratio at the same nominal thickness of 1.5 nm. Moreover, structures grown at a higher Ga/N_2_* ratio exhibited a stronger localization effect. In addition, MQW structures grown on different AlN/c-Al_2_O_3_ templates fabricated using either PA MBE or a combination of MOCVD and PA MBE showed similar PL spectra, despite some differences in the density of the growth spirals. The samples with the brightest PL peak intensities were selected for the further study of their CL spectra, which are described in the following paragraphs.

#### 3.4.2. Cathodoluminescence (CL) Spectra

The initial measurements of CL spectra were performed using conventional low-current e-guns with thermionic cathodes, which provided continuous e-beam excitations with a maximum current of up to 1 mA. The low-current CL spectra shown in Figure 9a exhibit the same features of the PL spectra described above in Section 3.4.1, i.e., a maximum red shift from 230–240 to 260–270 nm, as well as the broadening of single CL peaks for the MQW structures grown at the higher Ga-rich conditions. This corresponds to the conclusions attained concerning larger lateral dimensions and size dispersions of the carrier localization regions, characteristic for more Ga-rich growth conditions.

In addition, the measurements of the spatial behavior of CL spectra along the radius of the MQW wafer, shown in Figure 9a, demonstrate a relatively small red shift of the CL peak position, with an increase in its intensity as the measurement position shifts from the center to the edge of the 2-inch wafer. This effect can be associated with a slight increase (by about 10%) in the ratio of Ga/N_2_* fluxes towards the edge of the substrate due to a decrease in the intensity of activated nitrogen flux in the radial distribution in our MBE setup [64]. Therefore, the observed increase in the CL intensity was similar to the dependences of the PL spectra on the Ga/N_2_* ratio discussed in the previous sections.

Similar to PL, the structures grown under the same conditions on different AlN templates showed shapes and behaviors similar to the CL spectra in Figure 9a. Moreover, the MQW structures showed the same spectral peak positions in CL excited by e-beams with 10 or 30 keV in Figure 9b, which corresponds to penetration depths of about 0.75 µm and more than 6 µm. This indicates the high vertical uniformity of the MQW structure throughout the entire thickness of about 2 µm, which was also revealed by the AFM data for the MQW structures with different thicknesses of 1750 and 440 nm, discussed in Section 3.3.1 (Figure 4g–j). 

Finally, the measurements of CL excited with different electron energies shed light on the origin of the long-wavelength CL emission at wavelengths of either 370 or 470 nm, which are only observed for excitation by 30 keV, where a high number of electrons can reach the underlying AlN. Thus, this luminescence originated from the underlying AlN layers. As candidates for low-energy lines, the typical luminescence bands associated with point defects can be considered, namely, Al vacancies [V_Al_] emitting at 2.7 eV [65] and their complexes with oxygen impurities [V_Al_-(O_N_)_n_] emitting in a broad band at 3.4–3.8 eV [66], substitutional carbon on the nitrogen site and a nitrogen vacancy [C_N_-V_N_] with a prominent 2.8 eV emission peak [67], etc. In addition, CL at 330 nm might be associated with the emission of F+ centers in the *c*-sapphire substrate, into which a high-energy electron beam can penetrate [68].

The high uniformity of the MQW structures in the growth direction is extremely important for the implementation of high-power UVC emitters with high-current e-beam pumping using e-guns with ferroelectric plasma cathodes [69,70]. Figure 10a shows the CL spectra of the MQW structures excited by an e-gun of this type at an e-beam energy of 12.5 keV under a pulse current up to 2 A. Figure 10b shows the temporal variation in the output optical power excited by a high-current e-beam with several amplitude peaks randomly distributed at an e-beam pulse duration of about 0.5 µs. By integrating the output-power pulse, the average energy of each pulse can be calculated to be about 5 µJ. The analysis of these spectra allowed one to plot the dependences of peak output optical power on the e-beam current, which are presented in Figure 10c. These power dependences are linear for all MQW structures within all ranges of output optical power variations, i.e., the droop regime was not yet attained.

Figure 10c shows that the highest pulse power up to 50 W at a wavelength of 267 nm was achieved for the Q1.5-2.2-D_1_ structure grown under highly metal-enriched conditions. The same MQW structures grown on the S-type template emitted power at almost the same wavelength of 265 nm, and demonstrated somewhat lower power values of about 35 W. The maximum power value of 10 W at a wavelength of 238 nm was observed for structures with 1.5 ML-thick MQWs grown under a low Ga/N flux ratio of 1.1 on both S- and D_2_–type AlN/*c*-Al_2_O_3_ templates.

## 4. Conclusions

In summary, we demonstrated the growth of 400 × {GaN_1.5_/AlN_16_} MQW structures with the same nominal fractional QW thickness of 1.5 ML by PA MBE on various types of AlN/*c*-Al_2_O_3_ templates prepared either with PA MBE only or with the sequential use of MOCVD and PA MBE. The absence of a Stranski–Krastanov transition in these structures grown under Ga-rich conditions was confirmed by a streaky RHEED pattern throughout the growth of QWs and barrier layers. The metal-rich growth conditions led to wide terraces (in AFM) that allowed the formation of ML-thick islands spreading less than the width of a terrace, as confirmed by XRD and STEM. The tensile stresses commonly observed in AlN/*c*-Al_2_O_3_ templates and MQW structures were eliminated in the structures grown on the PA MBE AlN/c-Al_2_O_3_ templates and significantly reduced in the templates grown sequentially by MOCVD and PA MBE, which made it possible to suppress cracking in the thick MQW structures.

The PL spectra showed significantly higher intensities by more than an order of magnitude in the spectral range of 230–265 nm for structures grown under Ga-rich conditions. The peak wavelength increased with increasing the Ga/N_2_* ratio up to 2.2 due to an increase in the average thickness of QDKs and their lateral size. Energy-dependent CL spectra at low-current e-beams showed emission behaviors similar to PL and revealed the high vertical value and lateral uniformity of the 400 × QW structures over a two-inch wafer. Finally, 400 × {GaN_1.5_/AlN_16_} structures grown at Ga/N_2_* ratios of 2.2 and 1.1 and excited by a high-current e-gun demonstrated the maximum output optical powers of 50 and 10 W for UVC radiation at wavelengths of 267 and 238 nm, respectively, as well as linear non-saturated power dependencies.

## Figures and Tables

**Figure 1 nanomaterials-13-01077-f001:**
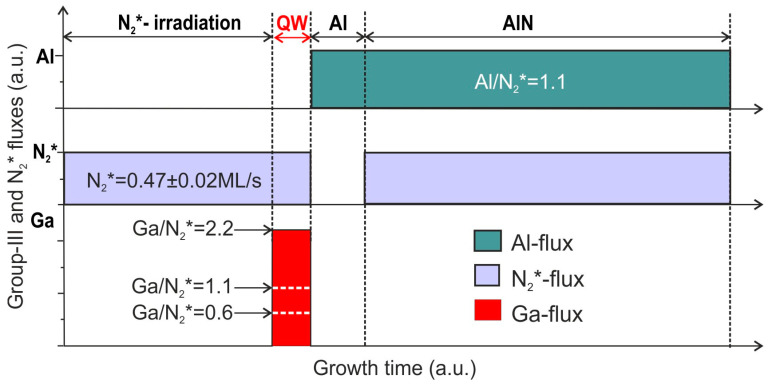
Scheme of the growth of GaN_1.5_/AlN_16_ QW structures at various Ga/N_2_* flux ratios determined by the Ga fluxes marked by dashed lines in their time diagrams.

**Figure 2 nanomaterials-13-01077-f002:**
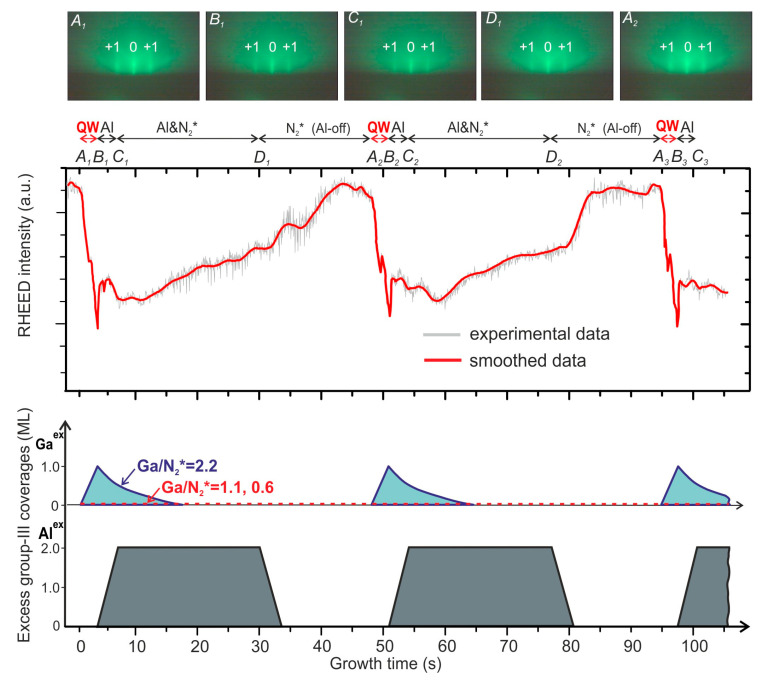
Time variation in the intensity of RHEED specular reflex (denoted in the photographs by the number 0) during the growth of two periods of 400 × {GaN_1.5_/AlN_16_} MQW structures under Ga-rich conditions (Q1.5-2.2-S). The upper insets show RHEED patterns recorded at different characteristic temporal points: *A*_i_, *B*_i_, *C*_i_, *D*_i_ (*i* = 1, 2). The lower insets show time diagrams illustrating the accumulation of excess Ga(Ga^ex^) and Al (Al^ex^) atoms during the growth of MQW structures with QWs grown at various Ga/N_2_* flux ratios. When plotting these diagrams, the Ga desorption rate was equal to 0.3 ML·s^−1^ and the desorption of Al was negligible.

**Figure 3 nanomaterials-13-01077-f003:**
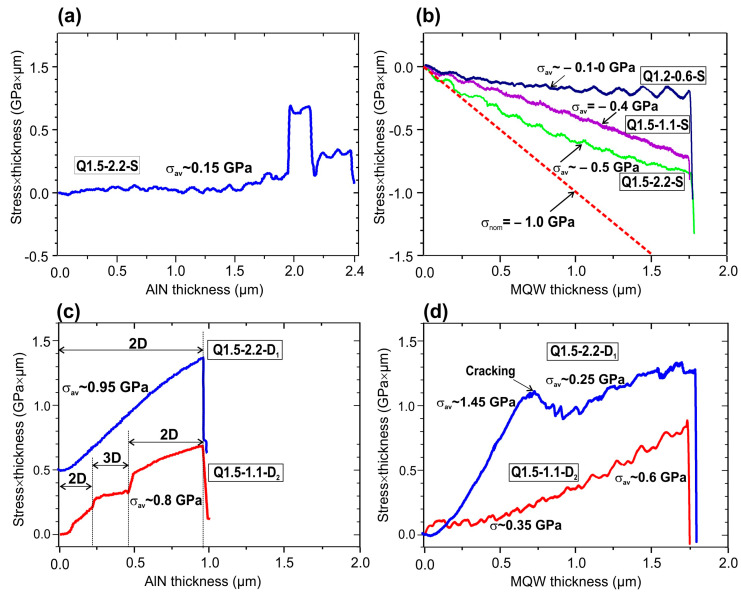
Products (stress × thickness) measured by MOSS during the PA MBE growth of: (**a**) a 2.4 µm-thick S-type AlN/*c*-Al_2_O_3_ template; (**b**) 400 × {GaN_1.5_/AlN_16_} MQW structures grown on S-type templates under various Ga/N_2_* flux ratios. The dashed, red line corresponds to product’s calculated stress × thickness in a film with a nominal compressive stress of −0.9 GPa; (**c**) two types of 0.9 µm-thick AlN templates D_1_ (blue line) and D_2_ (red line) (the former line is shifted upwards for clarity). The thicknesses of parts of the D_2_ template grown in 2D- and 3D-growth modes are shown via the lower, red curve; (**d**) 400 × {GaN_1.5_/AlN_16_} MQW structures Q1.5-2.2-D_1_ (blue line) and Q1.5-1.1-D_2_ (red line). The marked thickness on the blue line in (**d**) corresponds to the occurrence of cracking in this structure.

**Figure 4 nanomaterials-13-01077-f004:**
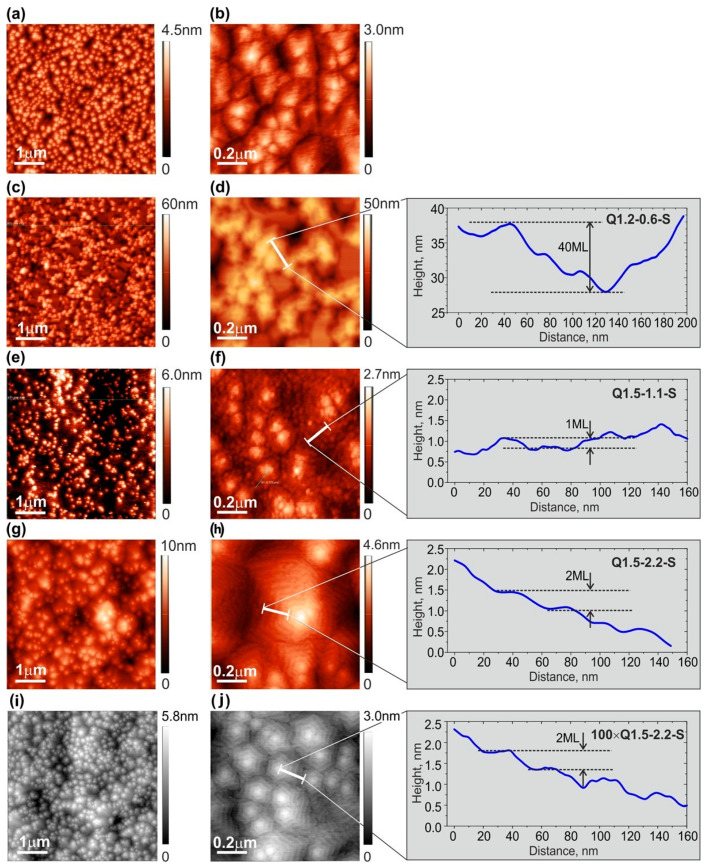
AFM images of the surfaces of a test AlN layer (**a**,**b**) and 400 × MQW structures with QWs grown at various Ga/N_2_* ratios: Q1.2-0.6-S (**c**,**d**); Q1.5-1.1-S (**e**,**f**); and Q1.5-2.2-S (**g**,**h**). AFM image of 100 × Q1.5-2.2-S structure grown under the same conditions as the previous one (**i**,**j**). AFM images have scanned areas of 1 × 1 µm^2^ and 5 × 5 µm^2^. Inserts in figures (**d**,**f**,**h**,**j**) depict changes in the profile height along the white line segments plotted on these figures.

**Figure 5 nanomaterials-13-01077-f005:**
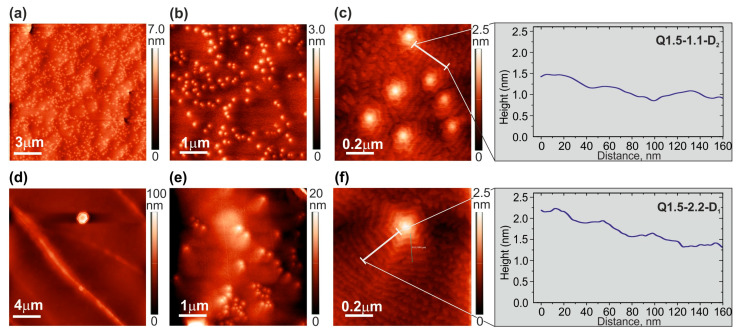
AFM images of the surfaces of MQW structures Q1.5-1.1-D_2_ (**a**–**c**) and Q1.5-2.2-D_1_ (**d**–**f**) grown on D_i_-type templates. The AFM images were measured with various scan areas ranging from 1 × 1 µm^2^ to 20 × 20 µm^2^. Inserts in figures (**c**,**d**) depict changes in the profile height along the white line segments plotted on these figures.

**Figure 6 nanomaterials-13-01077-f006:**
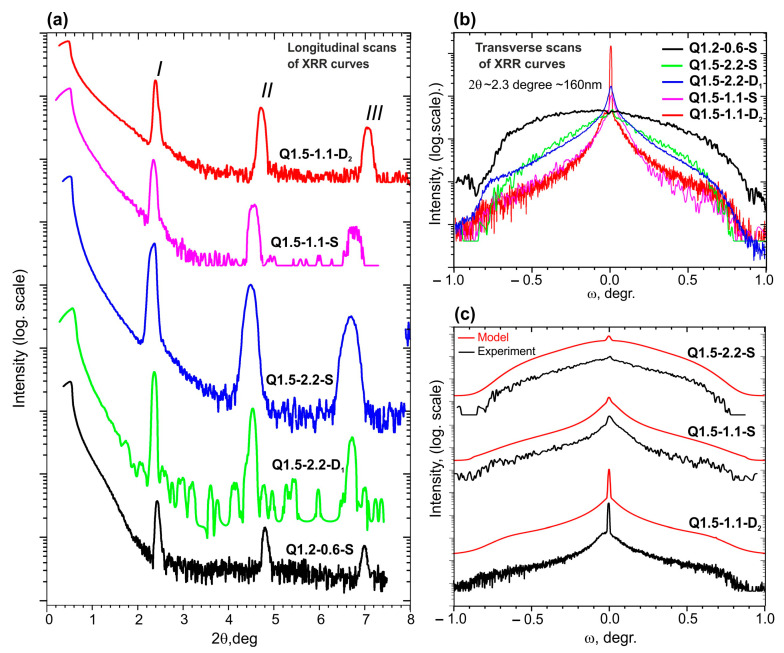
(**a**) X-ray reflectivity curves of 400 × {GaN_m_/AlN_16_} (*m* = 1.2, 1.5) MQW structures; (**b**) ω scans of the first Bragg-like peaks of the 400 × {GaN_m_/AlN_16_} (*m* = 1.2, 1.5) MQW structures. (**c**) Experimental (black) and simulated (red) ω scans of the first Bragg-like peaks of 400 × {GaN_1.5_/AlN_16_} MQW structures.

**Figure 7 nanomaterials-13-01077-f007:**
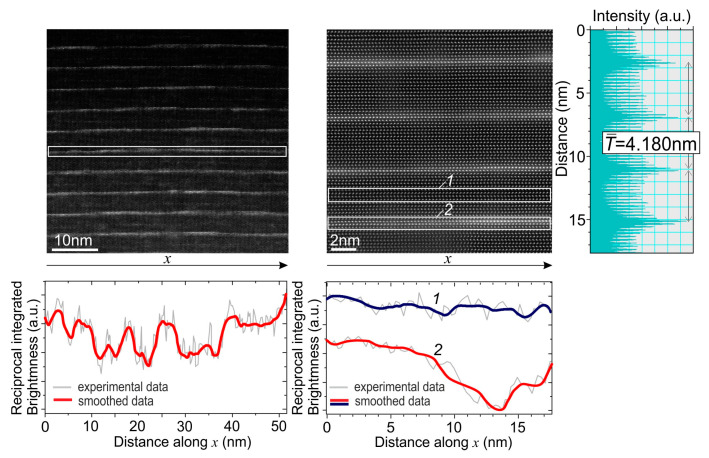
HAADF STEM images of the Q1.5-2.2-S structure taken at various magnifications. The right inset shows the integrated brightness distribution in the vertical direction of the adjacent image. The profiles plotted under each image show the spatial distribution in the lateral direction of the reciprocal values of the local brightness in the images at each point along the x-axis, integrated vertically within the white frames shown in the images: *1*—between the QWs; *2*—around the QW.

**Figure 8 nanomaterials-13-01077-f008:**
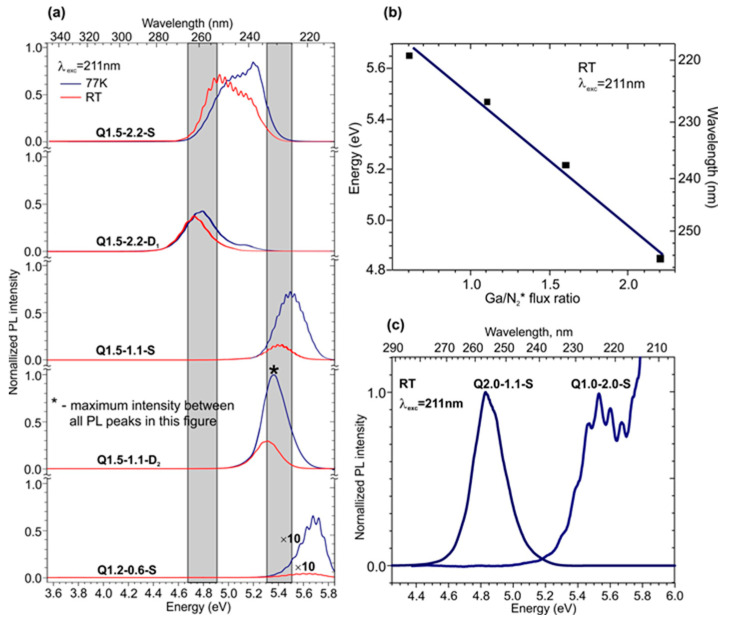
(**a**) PL spectra measured at RT (red) and 77 K (blue) in 400 × {GaN_m_/AlN_16_} (*m* = 1.2, 1.5) structures grown under various Ga/N_2_* flux ratios on different AlN templates. The right and left, gray areas show the energy positions of the PL peaks in MQW structures with integer QW thicknesses of 1 and 2 ML. (**b**) Dependence of the spectral position of the PL peak (RT) on the Ga/N_2_* flux ratio used to grow 400 × {GaN_1.5_/AlN_16_} structures on the S-type template. (**c**) PL spectra measured at RT in the 400 × {GaN_m_/AlN_16_} MQW structures with various nominal integer thicknesses of the wells (*m* = 1 and 2) and grown at a different Ga/N_2_* ratio on the S-type template.

**Figure 9 nanomaterials-13-01077-f009:**
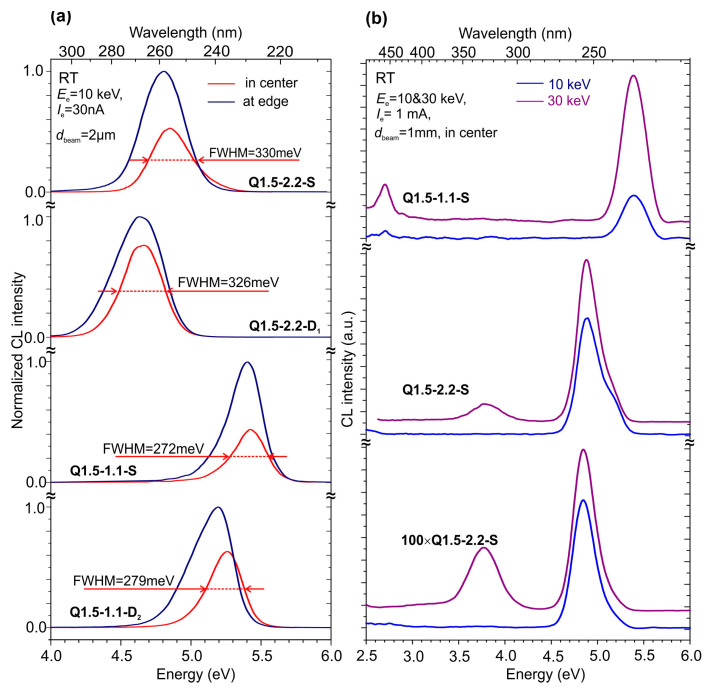
(**a**) Normalized CL spectra at RT excited with a “low-current” thermionic e-gun at the same *E*_e_ = 10 keV *I*_e_ = 30 nA, which were measured in the center (red) and at the edge (blue) of a 2-inch substrate in 400×{GaN_1.5_/AlN_16_} MQW structures grown under various Ga/N_2_* flux ratios on different AlN templates. (**b**) CL spectra excited at RT using a “low-current” thermionic e-gun with *E*_e_ = 10 (blue) and 30 (violet) keV, *I*_e_ = 5 µA, which were measured in 400 × {GaN_1.5_/AlN_16_} MQW structures grown under various Ga/N_2_* flux ratios on different AlN templates.

**Figure 10 nanomaterials-13-01077-f010:**
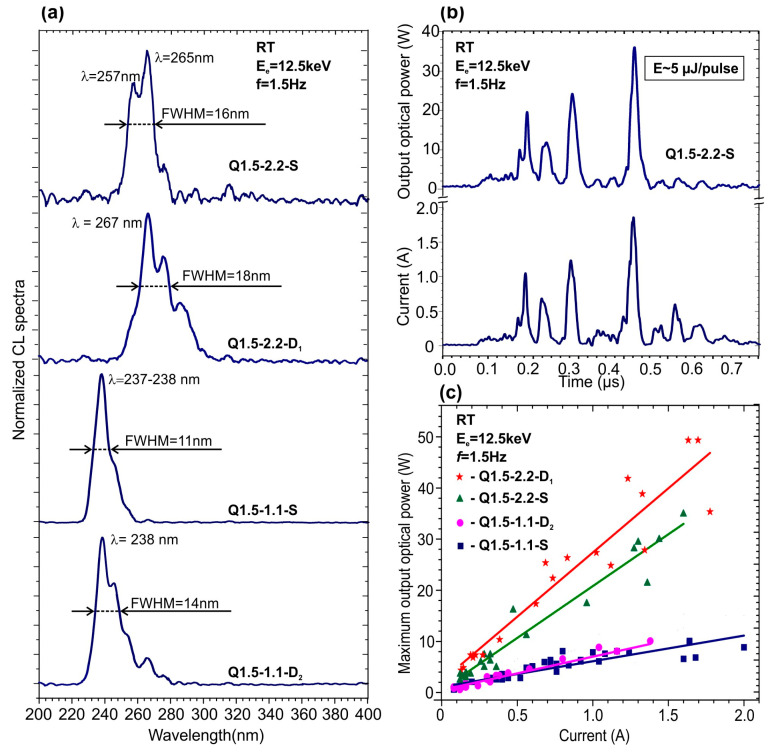
(**a**) Normalized CL spectra at RT excited with a “high-current” pulse e-gun at the same *E*_e_ = 12.5 keV, which were measured in 400×{GaN_1.5_/AlN_16_} MQW structures grown under various Ga/N_2_* flux ratios on different AlN templates. (**b**) The output optical power (upper, blue) from structure Q1.5-2.2-S excited by the e-beam current supplied by a “high-current” pulse e-gun at *E*_e_=12.5 keV, the time changes of which are shown in the lower violet curve. (**c**) Dependences of the output optical power on the current of an e-beam with an energy of 12.5 keV, measured for different 400 × {GaN_1.5_/AlN_16_} MQW structures. The lines are drawn in accordance with the linear interpolation of the experimental data.

**Table 1 nanomaterials-13-01077-t001:** Characteristics of surface topographies for the AlN layer and MQW structures.

	AlN Test Layer			Q1.2-0.6-S			Q1.5-1.1-S			Q1.5-2.2-S			Q1.5-1.1-D_2_			Q1.5-2.2-D_1_		
	Scanning Area *			Scanning Area			Scanning Area			Scanning Area			Scanning Area			Scanning Area		
	A	B	C	A	B	C	A	B	C	A	B	C	A	B	C	A	B	C
RMS, nm	0.54	0.59	0.43	8.1	8.2	9.3	0.57	0.51	0.35	1.27	1.27	0.79	0.51	0.34	0.31	10.2	0.67	0.20

* AFM scanning areas are A—15 × 15 µm^2^; B—5 × 5 µm^2^; C—1 × 1 µm^2^.

**Table 2 nanomaterials-13-01077-t002:** Parameters of MQW structures found from the simulations of data measured by XRD θ/2θ scans and longitudinal scans of XRR.

# Sample	Average Al Content,		MQW Period		RMS Surface
	Mol.%		nm		nm
	Nominal	XRD	Nominal	XRR	XRR
Q1.2-0.6-S	93.0	98.6	4.269	3.69 ± 0.3	>>1.0
100 × Q1.5-2.2-S	91.1	93.9	4.373	4.30 ± 0.12	1.1
Q1.5-2.2-S	91.1	96.5	4.373	4.00 ± 0.08	0.8
Q1.5-1.1-S	91.1	96.9	4.373	4.267 ± 0.04	0.6
Q1.5-2.2-D_1_	91.1	96.1	4.373	4.179 ± 0.035	0.5
Q1.5-1.1-D_2_	91.1	96.1	4.373	3.894 ± 0.008	0.3

**Table 3 nanomaterials-13-01077-t003:** Roughness and lateral and vertical correlation lengths; Hurst parameters of MQW structures from the simulation of transverse XRR scans of the first Bragg reflection in Figure 6c.

№ Sample	Roughness Total *	Correlation Lengths		Hurst Parameter
		Lateral	Vertical	
	nm	nm	nm	
Q1.5-1.1-D_2_	0.3	>200	-	0.3
Q1.5-1.1-S	0.6	>200	-	0.6
Q1.5-2.2-S	0.8	>100	4.0	1.0

*—calculated for both the surfaces and interfaces.

## Data Availability

The data presented in this study are available on request from the corresponding author.

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
