# Peer review of "2D-GaN/AlN Multiple Quantum Disks/Quantum Well Heterostructures for High-Power Electron-Beam Pumped UVC Emitters"

_nanomaterials, 2023, doi:10.3390/nano13061077_

Round 1

Reviewer 1 Report

The authors provided a very detailed and timely study of the AlN/GaN heterostructures for UVC emitters. Precise control over the thickness of the well and barrier layers is achieved via MBE and verified by TEM. The growth controllability and the light emission performance are of importance to the community. I thus suggest the manuscript be accepted as is.

Author Response

Thank you for your high appreciation of our work.

Reviewer 2 Report

The article reports on the 2D growth of GaN on AlN to create quantum films with a thickness of less than two monolayers. The authors describe the growth procedure as well as the in-situ characterisation by RHEED varying the Ga to N ratio. A analysis of the surface morphology as well as crystal quality characterisation by XRD is provided. Growing 400 GaN layers in sequence the authors demonstrate the high crystal quality and a 2D growth with manifests in rather large GaN islands denoted as quantum disks. The authors demonstrate the high crystal quality by PL and CL characterisation and demonstrate that the emission wavelength can be tuned by the controlling the growth conditions.

The article is comprehensible and provides the information to follow up the experiments. The authors provide a detailed analysis and the experimental data confirms the conclusion of the article. While the groth of GaN/AlN may not be considered as a novelty the paper provides information which are of relevance and interest in the field of III-nitride optoelectronic device fabrication. Publication is recommended.

Author Response

Dear Reviewer,

Thank you for the positive evaluation of our work. We agree that the growth method is not described in great detail, but this work is a continuation of our previous work Ref.[24], in which these methods were described in detail.

Reviewer 3 Report

This paper presents comprehensive study of a series of GaN/AlN multi-quantum well/disk structures that can be used for UVC e-beam pumped emitters. A variety of characterization methods including AFM, XRD, TEM, etc., are applied to study the samples’ structural properties and they show good agreement. The optical properties are studied by PL and CL, demonstrating single band emissions with a wide spectral range and different intensities and linewidths. This work is of the community’s interest. The paper has great discussions. However, there are several critical issues that the authors should address before this work can be recommended for publication. 

1. (line 51) It is not accurate to claim that the use of e-beam pumped emitters can bypass UV LED’s p-doping problem, because they are not direct replacements of UV LEDs. Please revise the phrasing. 

2. (line 115) Please provide details of the nitrogen plasma power and gas flow. These parameters might not be directly transferrable but are still useful reference for readers if they want to replicate the results. 

3. (line 176) The meaning of “kinetically restrict the stress relaxation” is not clear. Please elaborate (limited surface diffusion? less dislocation generation?) This statement may not be necessary and can be removed if it is not supported by evidence. In my opinion, Metal/N flux ratio is the dominating factor in controlling adatom surface diffusion and thus suppressing the 3D growth mode. The surface diffusivity provided by the metal bilayer in high metal/N ratio growth is an order of magnitude higher than the one that is thermally activated due to the low metal-metal binding energy. 

4. (Figure 2) Please specify from which RHEED streak was the intensity measured. 

5. (Section 3.2) I am curious about how much threading dislocations are playing a role in the discussion about stress generation/relaxation. Please comment. In addition, for the 3 types of AlN templates, can the author provide basic characterizations that are deemed necessary by the authors (e.g., HRXRD rocking curves, cross-sectional TEM images, AFM images, etc.)? They can be critical information for the readers because they lead to key results in this work. 

6. (line 285) Related to comment #3, the Al/N ratio of 1.1 is not particularly high, which, what I tend to believe, is the reason of the reduced surface adatom mobility, instead of the low T. 

7. (Figure 4 and 5) The lower bounds of the height-color profile are missing in all the AFM images. 

8. (Figure 6) Please either label the curves with sample numbers in Figure 6a and 6b like the ones in Figure 6c, or add legend if the Figure is too crowded. It is difficult to read from the caption. 

9. (Section 3.4.1) My general comment for this section is that it is unclear for the readers why these 5 particular samples are selected for PL demonstration in Figure 8 out of many possible combinations. Ideally, I would like to see the PL results of each one of the samples among Q[1.5,1.2]-[2.2,1.1,0.6]-[S,D1,D2] all 18 combinations, which I understand is not realistic. However, please educate the readers why the 5 samples are chosen to be presented. 

10. (Figure 8b, line 542) What are the 4 samples used to draw this plot? The only variable should be the Ga/N flux ratio among these 4 samples. Please confirm that the AlN template and all the other parameters are the same for all 4 samples. Otherwise, it is not scientifically rigorous to make such plot. 

11. (Section 3.4.1 and 3.4.2) For both PL and CL discussion, I would like to raise the question on dislocations again. Could the author comment on the effects of dislocations? For example, The author highlighted the difference in AlN templates. I would like to know the dislocation density of the starting AlN templates. They can certainly be extended into the MQW structures and become non-radiative recombination centers that affects light emission. 

Author Response

Dear Reviewer,

We are thankful to you for your careful reading, as well as for comments and suggestions that eliminate some errors and made our manuscript clearer for readers. We tried to address all the issues pointed out by you and correct the manuscript in accordance.

  1. (line 51) It is not accurate to claim that the use of e-beam pumped emitters can bypass UV LED’s p-doping problem, because they are not direct replacements of UV LEDs. Please revise the phrasing. 

We can formulate this idea as follows

It is generally accepted that the deterioration of p-type doping of AlGaN layers with increasing Al content is one of the most urgent problems of UVC LEDs. Therefore, it is of great interest to develop alternative electron-beam (e-beam) pumped UVC emitters (also called UVC-light-source tubes) based on (Al,Ga)N-based heterostructures with multiple quantum wells (MQW) without p-type doped layers [9-17]. Moreover, these UVC emitters, have demonstrated uniquely high output optical power up to the Watt range.

  1. (line 115) Please provide details of the nitrogen plasma power and gas flow. These parameters might not be directly transferrable but are still useful reference for readers if they want to replicate the results. 

New edition with added information:

The MQW structures were grown at the constant substrate temperature of 690°С and plasma-activated nitrogen flux N2*=0.47±0.02 ML s-1, which was provided by the plasma source excited by a RF-power of 150 W and a neutral nitrogen flux of 4 sccm. All MQW samples were grown with the same nominal QW thickness of 1.5 ML, but at various Ga/N2* flux ratio by changing of the Ga flux, as shown in Figure 1.

  1. (line 176) The meaning of “kinetically restrict the stress relaxation” is not clear. Please elaborate (limited surface diffusion? less dislocation generation?) This statement may not be necessary and can be removed if it is not supported by evidence. In my opinion, Metal/N flux ratio is the dominating factor in controlling adatom surface diffusion and thus suppressing the 3D growth mode. The surface diffusivity provided by the metal bilayer in high metal/N ratio growth is an order of magnitude higher than the one that is thermally activated due to the low metal-metal binding energy

We agree that statement “kinetic restriction the stress relaxation” can be unclear without a detailed analysis of the works [38-41] on growth of GaN/AlN QW structures over wide ranges of thickness, Ga/N flux ratio, growth temperature. Therefore, we remove this statement, but added important ones on ability of PA MBE to provide 2D growth of ultra-thin GaN QW (up to 2ML) in AlN at different Ga/N flux ratio (0.8-1.6) [38,39] and substrate temperature (690-740°C [40]). It should be added that Ga-bilayer did not during the growth of ultra-thin QWs. In accordance with the estimations presented in Supplementary 1, the growth of 1.5-ML-thick QW was accompanied by the formation of excess Ga with a maximum thickness of only 1ML at Ga/N=2.2 and was completely absent at lower Ga/N=1.1, 0.8.

Thus, this paragraph can be replaced with the following:

The main target of this work is fabrication of ultrathin GaN/AlN QWs with the well thickness of 1.5 ML (0.37 nm) which does not exceed the critical one of the Stranski-Krastanov transition in this heterostructure, reported as above 2 ML by different groups [38-41]. Moreover, these works show that PA MBE can provide a 2D growth mode of the ultra-thin (up to 2ML) GaN QWs grown in AlN over a wide range of Ga/N flux ratio (0.8-1.6) and substrate temperature (690-740°C). In addition, low growth temperature of about 690°C prevents the segregation and interdiffusion of Ga atoms at the GaN/AlN heterointerfaces [42].

At present, we are preparing the paper with detailed analysis of the growth kinetics of ultra-thin GaN QWs in AlN. Thank you for valuable comment.

  1. (Figure 2) Please specify from which RHEED streak was the intensity measured

Information about the designation of the RHEED specular reflex is added to the caption to Figure 2:

Figure 2. Time variation of the intensity of RHEED specular reflex (denoted in the photographs by the number 0) during the growth of two periods.

  1. (Section 3.2) I am curious about how much threading dislocations are playing a role in the discussion about stress generation/relaxation. Please comment. In addition, for the 3 types of AlN templates, can the author provide basic characterizations that are deemed necessary by the authors (e.g., HRXRD rocking curves, cross-sectional TEM images, AFM images, etc.)? They can be critical information for the readers because they lead to key results in this work. 

Answering the question about the role of threading dislocations in the generation/relaxation of elastic stresses in AlN hexagonal buffer layers and MQW structures, we do not think about the significant influence of this factor. Indeed, according to the generally accepted theory, formulated by Romanov&Speck in Ref. [Appl.Phys.Lett. 83,2003], the TDs in (0001) oriented layers usually have pure edge character with (0001) line direction and Burgers vector in the basal plane. The (1-100) prismatic glide planes of these dislocations are normal to the biaxial stress plane and there is no shear stress in the glide planes, which eliminates the contribution of TDs in stress change during layer growth.

Alternative effective mechanisms of stress evolution in AlN/c-Al2O3 templates during their PA MBE growth are described in detail in our previous work Ref. [43]. The near-zero stress observed in this work in the 2.4-µm-thick AlN/c-Al2O3 template fully correspond to the results of this work. TEM, AFM images and HRXRD rocking curves of this type of the AlN/Al2O3 templates are presented and discussed in our previous works Refs.[34,43].

An analysis of the stress evolution in ML-thick MQW GaN/AlN structures grown on PA MBE AlN/c-Al2O3 templates was started in our previous paper [24], where we found a significant difference between the observed near-zero stress and the expected compressive stress due to lattice constant mismatch in the MQW/AlN heterostructure. This work generally confirmed this tendency and will be continued in future works.

In the opposite case of the MOCVD-grown AlN/c-Al2O3 template, the observed tensile stress is fully consistent with conventional Hoffman-Nix-Clemens model explaining this stress by coalescence of growth grain during high temperature MOCVD growth, as described by Radhavan and Redwing in Ref.[44]. In our work, we observed the inheritance of this stress both in the AlN-PA MBE layer and followed GaN/AlN MQW structure, which can lead to structure cracking. The main result of our work is the proposed method for reducing tensile stresses in the AlN-PA MBE layer by a short-term transition to the 3D growth mode.

Finally, in this paper, we have focused on the study of the formation of ML-GaN/AlN MQW structures via the spiral growth and/or two-dimensional nucleation growth mechanisms. It is well known that screw dislocations determine the density of growth spirals in layers (heterostructures) (Ref.48). Therefore, we studied the influence of these dislocations on the surface topography of MQW structures grown on various AlN/c-Al2O3 templates. On page 9 (lines 347-348) we give significantly different half-widths of symmetric (0002) AlN XRD rocking curves for both types of templates, which is associated with a significantly lower density of screw dislocations in templates grown by MCVD. This corresponds to a noticeable difference in the surface topography of the MQW structures grown on these types of templates, as shown in Fig. 4 and 5.

  1. (line 285) Related to comment #3, the Al/N ratio of 1.1 is not particularly high, which, what I tend to believe, is the reason of the reduced surface adatom mobility, instead of the low T. 

Figure 1 shows that the GaN QWs are covered by Al-bilayer before the growth of each AlN barrier layers that results in their continuous growth at the increased adatom’s surface mobility even at relatively low values of both growth temperature (690ºC) and Al/N flux ratio (1.1).

In our previous paper [34] we have studied the effects of Al/N flux ratio (1.06-2.1), growth temperature and other growth parameters on the surface topography, threading dislocation densities and elastic stress evolution in AlN layers grown by metal-modulated epitaxy method in PA MBE. These studies experimentally revealed the crucial influence of Al-bilayer formation on the achievement of 2D surface topography of AlN layers. Therefore, since we provided a continuous coverage of the growing AlN layer with an Al-bilayer, the low growth temperature played the main role in the insufficient surface diffusion of adatoms .

  1. (Figure 4 and 5) The lower bounds of the height-color profile are missing in all the AFM images. 

The lower bounds of the height-color profile are added in all the AFM images. 

  1. (Figure 6) Please either label the curves with sample numbers in Figure 6a and 6b like the ones in Figure 6c, or add legend if the Figure is too crowded. It is difficult to read from the caption. 

This figure has been modified.

  1. (Section 3.4.1) My general comment for this section is that it is unclear for the readers why these 5 particular samples are selected for PL demonstration in Figure 8 out of many possible combinations. Ideally, I would like to see the PL results of each one of the samples among Q[1.5,1.2]-[2.2,1.1,0.6]-[S,D1,D2] all 18 combinations, which I understand is not realistic. However, please educate the readers why the 5 samples are chosen to be presented. 

This article stem from our previous investigations for similar structures over a much wider range of QW thicknesses and hence a wider spectral range from 3.1 to 5.25 eV [24]. In this work, we use a new technological method for the formation of such structures, described in Section 2, which allowed us to improve the reproducibility of the results and for the first time to study the dependence of the structural and optical characteristics of GaN/AlN MQW heterostructures on the Ga/N2* flux ratio. Therefore, in this section, we primarily compared samples grown at different values of this parameter. In addition, for the first time we have studied the structures grown on various AlN templates.

Moreover, the PL measurements helped us choose the brightest structures for further studies of the cathodoluminescence spectra which are also described in this article. Moreover, these studies helped us to choose the most promising directions for further research that are currently underway.

Thus, the following conclusion can be added in the end of this paragraph (Page 16, line 611):

Thus, in this section, we compared the PL spectra of ML thick GaN/AlN structures grown at different Ga/N2* flux ratios, which resulted in different QW growth mechanisms, as shown in 3.1.1. It was found that the position of the PL peaks can be controlled in the spectral range from 230 to 260 nm (4.7-5.5 eV) by increasing the Ga/N2* flux ratio at the same nominal thickness of 1.5 nm. Moreover, structures grown at a higher Ga/N2* ratio exhibited a stronger localization effect. In addition, MQW structures grown on different AlN/c-Al2O3 templates fabricated using either PA MBE or a combination of MOCVD and PA MBE showed similar PL spectra, despite some difference in the density of growth spirals. The samples with the brightest PL peak intensity were selected for further study of their CL spectra, which are described in the following paragraphs.

  1. (Figure 8b, line 542) What are the 4 samples used to draw this plot? The only variable should be the Ga/N flux ratio among these 4 samples. Please confirm that the AlN template and all the other parameters are the same for all 4 samples. Otherwise, it is not scientifically rigorous to make such plot

The following information has been added in the caption of Figure 8b :

Dependence of the spectral position of the PL peak (RT) on the Ga/N2* flux ratio used to grow 400´{GaN1.5/AlN16} structures on the S-type template.

  1. (Section 3.4.1 and 3.4.2) For both PL and CL discussion, I would like to raise the question on dislocations again. Could the author comment on the effects of dislocations? For example, The author highlighted the difference in AlN templates. I would like to know the dislocation density of the starting AlN templates. They can certainly be extended into the MQW structures and become non-radiative recombination centers that affects light emission. 

We fully agree with the reviewer that threading dislocations play an important role in the quantum efficiency and output optical power of any UVC emitter. Moreover, in the Introduction (pp. 1-2, lines 45-49) we demonstrate the best results for UVC LEDs, which were achieved on a bulk AlN substrate with a minimum threading dislocation density. One of the most important results of this work is the demonstration of UVC electron beam emitters with an output optical power that is significantly higher than the level achieved with any UVC LEDs and other types of e-beam pumped UVC emitters described in Refs.[4-17].

In our previous works [22,23,24], we demonstrated a continuous progress in the output parameters of e-beam pumped UVC emitters based on ML-thick GaN/AlN MQW structures. The problems of TD reduction and stress control in AlN/c-Al2O3 templates grown by PA MBE were also described in our separate works [34, 43]. In this work we used the best design of these templates, as well as standard MOCVD-grown AlN templates. Note that we focused on studying the growth kinetics of ML-thick GaN/AlN MQW structures and the demonstrating progress of the UVC output power level and the ability to control its wavelength are based on the results of these studies. Unfortunately, the standard volume of the article did not allow us to describe the effect of threading dislocations on the characteristics of UVС radiation of these structures, as well as the possibility of using the localization effect, the exciton nature of UVC radiation in ML-thick MQW structure and other effects to reduce the harmful effect of TD and improve the results of this work. This work will be continued.

Round 2

Reviewer 3 Report

I would like to thank the authors for the detailed response and fruitful discussion. I recommend this paper for publication. Great work. 

Please make sure the revisions are incorporated in the manuscript and proofread before final submission. 

1. The authors' response to comment #2 is not reflected in the latest version of the manuscript. Please insert the revised text at line 114. In addition, in the revised paragraph, "nitrogen flux of 4 sccm" should be "nitrogen flow of 4 sccm". 

2. The authors' response to comment #10 is not reflected in the latest version. Please revise accordingly. 

Author Response

Dear Reviewer,

Thank you for your careful reading and we've made the final corrections recommended:

  1. The authors' response to comment #2 is not reflected in the latest version of the manuscript. Please insert the revised text at line 114. In addition, in the revised paragraph, "nitrogen flux of 4 sccm" should be "nitrogen flow of 4 sccm". 

We inserted this paragraph.

  1. The authors' response to comment #10 is not reflected in the latest version. Please revise accordingly. 

We inserted this words in the caption of figure 8.

Sorry for my forgetfulness.